# Genomic characterization of *Staphylococcus aureus* isolates causing osteoarticular infections in otherwise healthy children

Walter Dehority[1]*, Valerie J. Morley[2], Daryl B. Domman[2], Seth M. Daly[3], Kathleen D. Triplett[3], Kylie Disch[1], Rebekkah Varjabedian[4], Aimee Yousey[5], Parisa Mortaji[6], Deirdre Hill[7], Olufunmilola Oyebamiji[8], Yan Guo[8], Kurt Schwalm[1], Pamela R. Hall[3], Darrell Dinwiddie[1], Jon Femling[5]

1 Department of Pediatrics, The University of New Mexico School of Medicine, Albuquerque, New Mexico, United States of America, 2 Department of Internal Medicine, The University of New Mexico School of Medicine, Center for Global Health, Albuquerque, New Mexico, United States of America, 3 Department of Pharmaceutical Sciences, The University of New Mexico College of Pharmacy, Albuquerque, New Mexico, United States of America, 4 The University of New Mexico, Albuquerque, New Mexico, United States of America, 5 Department of Emergency Medicine, The University of New Mexico School of Medicine, Albuquerque, New Mexico, United States of America, 6 Department of Internal Medicine, The University of Colorado School of Medicine, Aurora, Colorado, United States of America, 7 The University of New Mexico Clinical and Translational Science Center, Albuquerque, New Mexico, United States of America, 8 Division of Molecular Medicine, The University of New Mexico, Albuquerque, New Mexico, United States of America

* WDehority@salud.unm.edu

**Data Availability Statement:** Data are posted in the public repository 'Dryad', doi:10.5061/dryad. 2280gb5vj.

## Abstract

### Background

Pediatric osteoarticular infections are commonly caused by *Staphylococcus aureus*. The contribution of *S. aureus* genomic variability to pathogenesis of these infections is poorly described.

### Methods

We prospectively enrolled 47 children over 3 1/2 years from whom *S. aureus* was isolated on culture—12 uninfected with skin colonization, 16 with skin abscesses, 19 with osteoarticular infections (four with septic arthritis, three with acute osteomyelitis, six with acute osteomyelitis and septic arthritis and six with chronic osteomyelitis). Isolates underwent whole genome sequencing, with assessment for 254 virulence genes and any mutations as well as creation of a phylogenetic tree. Finally, isolates were compared for their ability to form static biofilms and compared to the genetic analysis.

### Results

No sequence types predominated amongst osteoarticular infections. Only genes involved in evasion of host immune defenses were more frequently carried by isolates from osteoarticular infections than from skin colonization (p = .02). Virulence gene mutations were only noted in 14 genes (three regulating biofilm formation) when comparing isolates from subjects with osteoarticular infections and those with skin colonization. Biofilm results

**Funding:** Funding: This work was supported by the National Center for Research Resources and the National Center for Advancing Translation Sciences of the National Institutes for Health through grant [2U54TR01449-06A1], the University of New Mexico Clinical and Translational Science Center and a University of New Mexico Departmental Research Allocation Committee award [WD]. This work was funded in part by NIAID NIH grant AI145324 to Pamela R Hall. The funders had no role in study design, data collection and analysis, decision to publish, or preparation of the manuscript.

**Competing interests:** The authors have declared that no competing interests exist.

demonstrated large heterogeneity in the isolates' capacity to form static biofilms, with healthy control isolates producing more robust biofilm formation.

## Conclusions

*S. aureus* causing osteoarticular infections are genetically heterogeneous, and more frequently harbor genes involved in immune evasion than less invasive isolates. However, virulence gene carriage overall is similar with infrequent mutations, suggesting that pathogenesis of *S. aureus* osteoarticular infections may be primarily regulated at transcriptional and/or translational levels.

## Introduction

Osteoarticular infections such as septic arthritis, acute osteomyelitis and chronic osteomyelitis may lead to severe sequelae in children, including limb length discrepancies [1, 2], suppurative complications [3], pathologic fractures [2], and a frequent need for surgery [3] and prolonged courses of antibiotics [4]. Though *Staphylococcus aureus* is the most common cause of osteoarticular infections in children [5–7], it is also a common cause of less invasive skin and soft tissue infections [8, 9] and also frequently colonizes the skin of asymptomatic individuals [9]. The genetic determinants of these various clinical phenotypes are not well understood. From a mechanistic standpoint, it would be anticipated that certain genes carried by *S. aureus* would play a unique role in osteoarticular infections. For example, the bone sialoprotein binding protein (a product of the *bbp* gene) is a bacterial cell wall protein that binds to matrix molecules on bone, antibodies to which correlate with the presence of osteomyelitis in adults [10], while mutations in the collagen binding adhesion gene (*cna*) in *S. aureus* render the organism significantly less able to produce osteomyelitis in a murine model of the disease [11]. However, the genetic regulatory processes involved in osteoarticular infections due to *S. aureus* are likely multifactorial, and may involve the genomic composition of an infecting isolate [12], transcriptional patterns of the organism [13], the influence of the host immune response and antibiotic exposure on the bacterial transcriptome [14–17] and the host microbiome [18–20], among others. Identification of the bacterial genetic processes involved in driving the predilection of a *S. aureus* isolate towards the formation of an osteoarticular infection, however, may provide potential targets for immunotherapeutic agents and vaccines [21], as well as novel diagnostic approaches to differentiate virulent from avirulent strains of the organism which may be colonizing a child's body. Previous work has emphasized murine models [14], single disease presentations (e.g. acute osteomyelitis only) [12, 15] or methicillin-resistant *S. aureus* [12, 15]. As a necessary first step in elucidating the genetic basis for *S. aureus* osteoarticular infections, we utilized whole genome sequencing to characterize the carriage of virulence genes in *S. aureus* isolates (methicillin susceptible and methicillin resistant) taken from children across a spectrum of disease, as well as to assess for mutations in these genes and to characterize the phylogenetic spectrum of these isolates. We compared results from children with acute osteomyelitis, chronic osteomyelitis and acute septic arthritis, as well as *S. aureus* isolates from children with skin abscesses and isolates collected from uninfected children with skin colonization. We hypothesized that the distribution of virulence genes would be similar across all phenotypes of disease, suggesting primary control of pathogenesis at the transcriptional and/or translational level.

## Materials and methods

### Construction of the study cohort

Subjects were prospectively enrolled between June 12th, 2016 and December 2nd, 2019. Subjects admitted to our hospital or Pediatric Emergency Department who were under 18 years of age and without known immune deficiencies or post-operative or orthopedic implant associated infections were eligible for enrollment. Subjects were enrolled from the following four groups of osteoarticular infections: 1.) acute osteomyelitis (symptoms <14 days, normal orthopedic plain films at admission and elevated inflammatory markers, as previously described) [22] 2.) acute septic arthritis (any subject requiring an arthrotomy for suspected septic arthritis with growth of *S. aureus* on culture of blood and/or synovial fluid) 3.) chronic osteomyelitis (symptoms >14 days at admission, abnormal orthopedic plain films at admission and histopathology supporting the diagnosis if available, with normal or mildly elevated inflammatory markers) 4.) concurrent acute septic arthritis and acute osteomyelitis. To better evaluate genomic composition across a spectrum of invasion, *S. aureus* isolates collected from two groups of controls were utilized: 1.) children with skin and soft tissue abscesses (with sterile blood cultures, if obtained, and no evidence of systemic invasion such as pneumonia or osteoarticular infections) and 2.) uninfected children with asymptomatic skin colonization who were admitted for non-infectious conditions (e.g. febrile seizures, asthma exacerbations). Demographic and clinical information were obtained for all subjects from the electronic medical record, save for healthy, uninfected controls who were promised anonymity.

### Microbiological methodology

For subjects with infection, bacterial isolates from clinical cultures were confirmed as *S. aureus* via matrix-assisted laser desorption time-of-flight (MALDI-TOF) analysis, and then collected from sub-cultures for sequencing. Multiple isolates may have been collected from the same subject (e.g. if cultures isolated *S. aureus* from multiple time points during admission), though genomic comparisons unless stated otherwise were based off the initial isolate. For uninfected control subjects, axillary or nasal swabs were collected and plated on mannitol salt agar. Coagulase-positive isolates fermenting mannitol underwent confirmatory MALDI-TOF analysis, and confirmed *S. aureus* isolates were saved for sequencing. All *S. aureus* isolates were frozen in 10% glycerol stock at -80 degrees until batched analysis. Susceptibility testing (to differentiate methicillin resistant *S. aureus*, MRSA, from methicillin susceptible *S. aureus*, MSSA) was performed with disk diffusion prior to freezing, and confirmed with molecular analysis for the *mecA* gene.

### Sequencing methodology

Prior to sequencing, isolates were re-cultivated in tryptic soy broth, mixed in DNA/RNA Shield lysis tubes™ (Zymo Research™) and centrifuged at 10,000 x g for 1 minute. DNA was isolated using the ZymoBIOMICS DNA isolation kit following the manufacturer's recommended protocol (Zymo Research™). More specifically, the resulting supernatant was added to a Zymospin™ filter, centrifuged at 8000 x g for 1 minute followed by addition of DNA binding buffer. The resulting mixture was added to a Zymospin™ column, centrifuged at 10,000 x g for 1 minute followed by rinses with DNA wash buffer. This was added to DNAse/RNAse free water and centrifuged at 10,000 x g for 1 minute. DNA was eluted from this using a Zymospin™ filter via centrifugation. The resulting DNA was prepared for Illumina next-generation sequencing using the Illumina Nextera XT DNA library prep kit, per recommended instructions. Completed sequencing libraries were assessed for quality and concentration by gel electrophoresis (Agilent™) and Qubit fluorometric quantitation (Thermo Fisher

Scientific™), respectively. Completed libraries were pooled in equimolar ratios and underwent whole genome sequencing via 2x250 bp sequencing using v3 sequencing reagents on an Illumina MiSeq (see S1 Table for number of sequencing reads).

## Bioinformatic and phylogenetic methodology

For analysis of virulence genes, FASTA sequences were identified for 254 virulence genes (genes taken from a published compilation [23] and a supplementary literature search). Sequencing data of *S. aureus* were aligned using BWA 0.7.17 [24] using *S. aureus* reference genome NCTC8325 downloaded from NCBI. Binary alignment map (BAM) files were sorted and indexed using Samtools 1.9 [25]. BCFTools 1.9 was used to count allele frequency from the BAM files. Transcriptome information of *S. aureus* was downloaded from GenBank as CP000253.1 general feature file and converted to gene transfer format (GTF) using GFF Utilities [26]. Then FeatureCounts [27] was used to count reads aligned to genes. Proportion tests were used to assess for a proportional difference of variants between case and control groups. Adjusted $p < 0.05$ was considered statistically significant.

For phylogenetic analysis, raw sequencing reads were trimmed with Trim Galore using default settings [28]. Assemblies were created with Unicycler (S1 Table) [29]. Sequence types were determined using ARIBA [30]. Forty-seven isolates (one from each patient) were included for phylogenetic analysis. For these 47 isolates, a core genome alignment was created with Roary [31]. A maximum likelihood phylogeny was built from the core genome alignment with IQ-TREE using 5000 ultrafast bootstraps and a GTR+G model of nucleotide substitution [32, 33]. Phylogenies were visualized using GGTREE [34]. Branches were analyzed by year, source of the sequenced isolate, the presence of the *mecA* gene and the type of infection. Given that the traditional classification of the types of osteoarticular infection as either septic arthritis, acute osteomyelitis or chronic osteomyelitis may be somewhat arbitrary and not reflective of a continuum of infection (e.g. both septic arthritis and chronic osteomyelitis may arise as complications of acute osteomyelitis), a severity of illness score was calculated for subjects with acute osteomyelitis as previously described [35] for assessment of phylogenetic relatedness and disease severity.

## Static biofilm assay

Static biofilm assays were conducted using a modified method of Cassat et al [36] that we recently described [37]. Briefly, 96-well plates were coated overnight at 4 ˚C with pooled human plasma (IPLANAC; Innovative Research, Novi, MI). Overnight cultures in duplicate for each strain were grown in TNB [trypticase soy broth (Becton, Dickinson and Company, Sparks, MD) with 0.5% w/v dextrose (VWR Analytical, Radnor, PA) and 3% w/v NaCl (Fisher Scientific, Waltham, MA)] at 37 ˚C with 220 rpm. Overnight cultures were $OD_{600nm}$ matched to within 0.05 and then diluted 1:200%v/v in fresh media. Coated wells were gently washed with phosphate buffered saline (PBS) and then inoculated with six technical replicates per biological replicate. PBS in coated wells served as a negative stain control. Plates were then incubated statically for 24 h at 37 ˚C. Non-adherent culture was aspirated, washed twice with PBS, and then and the wells were fixed with 100% v/v ethanol. Ethanol was removed and the plate was allowed to dry for 10 min. Biofilm was stained with 0.1% w/v crystal violet (Sigma-Aldrich) for 2 minutes and then aspirated and washed twice with PBS. The stain was eluted with 100% v/v ethanol by shaking for 10min and then diluted 1:10 in 100% v/v ethanol for $OD_{595nm}$ measurement. A USA300 *S. aureus* MRSA isolate (AH1263) and it's isogenic *agr*-deletion mutant (AH1292) were included on each plate as internal controls and biofilm comparators [37].

## Statistical methodology

Descriptive statistics including counts and frequencies were used to profile participant characteristics, including type of osteoarticular infection. For categorical variables, chi-square tests were calculated using Fisher's exact test for cell sizes less than five. For continuous variables, means, medians and interquartile range (IQR) were assessed. In addition to analysis of the distribution of individual genes between types of infection and controls, genes were also grouped into families according to putative function (toxin, adhesins, antibiotic resistance, immune evasion, proteases, hemolysins/leukocidins/hyaluronidases) as described in the literature, and the mean proportion positive for each family was calculated. Assignment of genes to a family was based upon putative functions listed on the website www.uniprot.org, a recently published review on the topic [23] and supplementary literature review. Differences in mean distribution between osteoarticular infections vs. healthy controls and vs. skin abscess controls were calculated using a t-test. Descriptive statistics were also employed to evaluate gene carriage in isolates from separate sources in the same patient (e.g. bone and blood cultures) and isolates from the same patient serially over time. All statistical tests were two-sided. To decrease the likelihood of false positive findings given the large number of statistical comparisons undertaken, the Benjamini and Hochberg correction [38] was used and reported as the final p-value. The quantity of biofilm production for each included bacterial isolate was compared between skin soft tissue infections, bone and joint infections and healthy controls, and to the biofilm comparators and internal controls AH1263 and to AH1292. Biofilm quantities were evaluated on a log scale to accommodate non-normal distributions. Mixed models were estimated with accounting for repeated measures. All statistical tests used a two-side alpha value of .05. Analyses were conducted using Statistical Analysis Systems (SAS) software, v. 9.4 (Cary, N.C.)

This study was approved by the University of New Mexico Human Research Review Committee (HRRC #16–102), and informed consent was obtained for all participants.

## Results

### Clinical, demographic and microbiological overview of the study population

Among uninfected controls, specimens for culture were collected from 127 subjects. Of these 127 subjects, 12 subjects produced specimens with growth of viable *S. aureus* isolates (11 from nasal swabs and one from an axillary swab). Six of these uninfected controls were enrolled from the inpatient units, and six from the Pediatric Emergency Department. Overall, sixteen subjects with skin and soft tissue abscesses were also enrolled, as well as four subjects with septic joints, three subjects with acute osteomyelitis, six subjects with both acute osteomyelitis and septic arthritis and six subjects with chronic osteomyelitis (Table 1). Subjects with osteoarticular infections were more likely to be male (p = .04) and were older (p = .02) (Table 1). MRSA was more frequently seen in isolates from skin abscesses (56.2%) than osteoarticular infections (15.8%), though this did not reach statistical significance (p = .06).

### Phylogenetic analyses of the isolates

To determine whether osteoarticular infections were caused by closely related bacterial strains, a phylogenetic tree was generated for 47 *S. aureus* isolates (Fig 1). Isolates causing osteoarticular infections were widely dispersed throughout the phylogeny, and were often closely related to isolates from skin infections and skin colonization. Sequenced isolates came from 11 different sequence types (ST8, ST15, ST789, ST5, ST87, ST59, ST51, ST30, ST39, ST398, ST45), and osteoarticular infections were observed from 7 sequence types (ST8, ST5, ST87, ST59, ST51,

**Table 1. Demographic, clinical and microbiological overview of 35 children with osteoarticular infections and skin and soft tissue abscesses.**

| Variable | SSTI | OAI | Exact p-value | Adjusted p-value |
|---|---|---|---|---|
| **Gender** | | | | |
| Male | 5 (31.3) | 15 (79.0) | .005 | .04 |
| Female | 11 (68.7) | 4 (21.0) | | |
| **Age (years)** | | | | |
| Mean | 1.3 | 11.1 | | |
| Median | 1.4 | 11.8 | | |
| IQR | 0.5–2.1 | 6.3–13.3 | .001 | .02 |
| **Race** | | | | |
| African American | 1 (6.3) | 0 | NA | NA |
| Asian | 1 (6.3) | 0 | | |
| Caucasian | 13 (81.2) | 17 (89.5) | | |
| Native American | 1 (6.3) | 2 (10.5) | | |
| **Ethnicity** | | | | |
| Hispanic | 5 (31.3) | 9 (47.4) | .33 | .48 |
| Non-Hispanic | 11 (68.7) | 10 (52.6) | | |
| **Frequency of MSSA** | 7 (43.8) | 16 (84.2) | .01 | .06 |
| **Frequency of MRSA** | 9 (56.2) | 3 (15.8) | | |

Due to rounding, numbers may not add to 100%

SSTI, skin and soft tissue infection; OAI, osteoarticular infection; IQR, intraquartile range; NA, not applicable; MSSA, methicillin susceptible *Staphylococcus aureus*; MRSA, methicillin resistant *Staphylococcus aureus*.

The 12 asymptomatic control subjects with skin colonization were promised anonymity, and did not contribute demographic information as a result.

ST30 and ST398). In other words, isolates causing invasive infections came from a diversity of lineages, and they were not phylogenetically distinct from less invasive isolates. No association was present between phylogenetic relatedness and severity of acute osteomyelitis.

ST8 was the serotype present in all but one of our isolates carrying the *mecA* gene, which in turn represented half of our skin and soft tissue abscesses, consistent with prior reports [39]. Isolates with the mecA gene accounted for all isolates with confirmed phenotypic antimicrobial resistance patterns consistent with MRSA.

## Distribution of virulence genes and virulence gene families

Immune evasion genes were significantly more common amongst *S. aureus* isolates collected from children with osteoarticular infections than from those collected from children with asymptomatic skin colonization after accounting for multiple comparisons (p = .02), but not when compared with isolates collected from children with skin abscesses (p = .76) (Tables 2 and 3, Fig 2). The absolute difference in carriage of these genes between isolates from osteoarticular infections and asymptomatic skin colonization was only 5%, however. The *Cap5h*, *Cap5l and Capj* genes individually were all significantly associated with osteoarticular infections in unadjusted analyses (p = 0.04), but this was not sustained when accounting for multiple comparisons.

## Spatial and temporal assessment of genomic content

Among *S. aureus* isolates taken from four subjects concurrently from different sources (three subjects with isolates collected from blood cultures and cultures of bone abscesses, and one subject with isolates collected from a blood culture and a synovial fluid culture) no differences

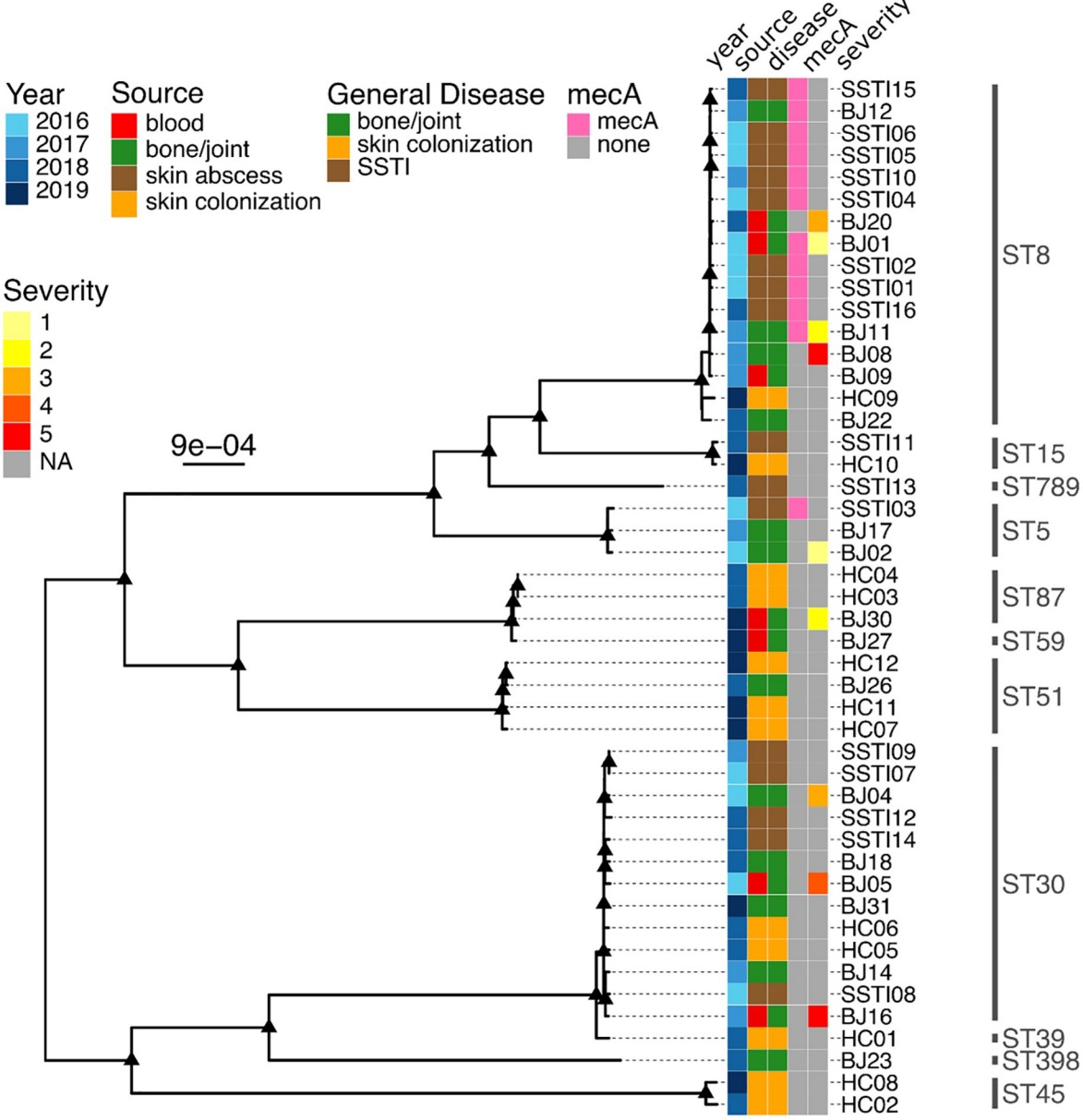

**Fig 1. Phylogenetic analysis of *Staphylococcus aureus* isolates from children with osteoarticular infections compared with children with skin and soft tissue abscesses and healthy controls.** Maximum likelihood phylogeny of 47 *S. aureus* genomes. Each isolate originates from a different patient. This phylogeny was generated from a core genome alignment, and it is midpoint rooted. Branches are color-coded by the year, source of the sequenced isolate, the patient's general disease, and the presence/absence of *mecA* and a disease severity score for patients with acute osteomyelitis (scale from 1, lowest to 10, highest). Sequence types are shown at the right. The scale bar indicates nucleotide substitutions per site. Triangles indicate nodes with ultrafast bootstrap support of at least 95%. BJ = bone and/or joint infection; SSTI = skin/soft tissue infection; HC = healthy control.

amongst carriage of virulence genes were seen in each individual subject's isolates. Among six subjects who had *S. aureus* isolates collected serially over time (four subjects from bone cultures three to seven days apart and two subjects with blood and synovial fluid cultures one to three days apart), no significant differences in virulence gene carriage were noted amongst each subject's isolates.

**Table 2. Distribution of virulence gene families amongst *Staphylococcus aureus* isolates from children with osteoarticular infections compared with healthy controls.**

| Gene Family | ASA | AOM | ASA+AOM | COM | OAI | Controls | p-value | Corr. p-value |
|---|---|---|---|---|---|---|---|---|
| Toxins | 0.50 | 0.54 | 0.54 | 0.51 | 0.52 | 0.55 | .28 | .46 |
| Adhesins | 0.92 | 0.90 | 0.90 | 0.92 | 0.91 | 0.94 | .13 | .41 |
| Antibiotic Resistance | 0.63 | 0.64 | 0.64 | 0.63 | 0.64 | 0.66 | .28 | .46 |
| Immune Evasion | 0.94 | 0.92 | 0.95 | 0.92 | 0.93 | 0.88 | .002 | .02* |
| Proteases | 0.63 | 0.55 | 0.86 | 0.75 | 0.73 | 0.55 | .26 | .46 |
| Hemolysins/Leukocidins/Hyaluronidases | 0.92 | 0.95 | 0.93 | 0.93 | 0.93 | 0.93 | .74 | .76 |

*denotes statistical significance after adjustment for multiple comparisons

ASA, acute septic arthritis (n = 3); AOM, acute osteomyelitis (n = 4); ASA+AOM, acute septic arthritis and acute osteomyelitis concurrently; COM, chronic osteomyelitis (n = 6); OAI, all osteoarticular infections combined (n = 19); Corr, corrected (via Benjamini-Hochberg correction of p-value)

Numbers in each column of osteoarticular infection represent the percentage of isolates carrying at least one of the genes belonging to each family.

## Distribution of virulence gene mutations

Mutations were most frequently documented when comparing sequences from acute osteoarticular infections (acute osteomyelitis or acute septic arthritis) and healthy controls (Fig 3). Most interestingly, significant differences were noted in the frequency of mutations of genes associated with biofilm formation and regulation, including the *clfA* gene (clumping factor A), the *clp* gene and the *WalKR* regulon. However, mutations were noted in only 14 virulence genes overall.

## Biofilm formation capacity

Biofilm formation is clinically relevant to osteoarticular infections and as such we sought to elucidate the isolate's ability to form biofilm in a 24-hour static biofilm assay. The isolates were normalized to USA300 LAC (AH1263) which is a clinical MRSA strain and compared with the isogenic *agr*-deletion mutant (AH1292). Forty-five of the 47 isolates were analyzed, as one isolate did not grow on culture, and another was not located in the freezer (both from asymptomatic skin colonization). There was heterogeneity within each infection type, where skin and soft tissue infection isolates ranged from 73%–284% (Fig 4A), osteoarticular isolates ranged from 59%–327% (Fig 4B), and healthy control isolates (asymptomatic skin colonization) ranged from 35%–295% (Fig 4C). Most surprisingly, when the isolates were grouped together the

**Table 3. Distribution of virulence gene families amongst *Staphylococcus aureus* isolates from children with osteoarticular infections compared with children with skin and soft tissue abscesses.**

| Gene Family | ASA | AOM | ASA+AOM | COM | OAI | Controls | p-value | Adj. p-value |
|---|---|---|---|---|---|---|---|---|
| Toxins | 0.50 | 0.54 | 0.54 | 0.51 | 0.52 | 0.52 | .88 | .88 |
| Adhesins | 0.92 | 0.90 | 0.90 | 0.92 | 0.91 | 0.91 | .76 | .76 |
| Antibiotic Resistance | 0.63 | 0.64 | 0.64 | 0.63 | 0.64 | 0.64 | 69 | .76 |
| Immune Evasion | 0.94 | 0.92 | 0.95 | 0.92 | 0.93 | 0.94 | .71 | .76 |
| Proteases | 0.63 | 0.55 | 0.86 | 0.75 | 0.73 | 0.84 | .15 | .41 |
| Hemolysins/Leukocidins/Hyaluronidases | 0.92 | 0.95 | 0.93 | 0.93 | 0.93 | 0.95 | .29 | .46 |

ASA, acute septic arthritis (n = 3); AOM, acute osteomyelitis (n = 4); ASA+AOM, acute septic arthritis and acute osteomyelitis concurrently; COM, chronic osteomyelitis (n = 6); OAI, all osteoarticular infections combined (n = 19); Adj, adjusted (via Benjamini-Hochberg correction)

Numbers in each column of osteoarticular infection represent the percentage of isolates carrying at least one of the genes belonging to each family.

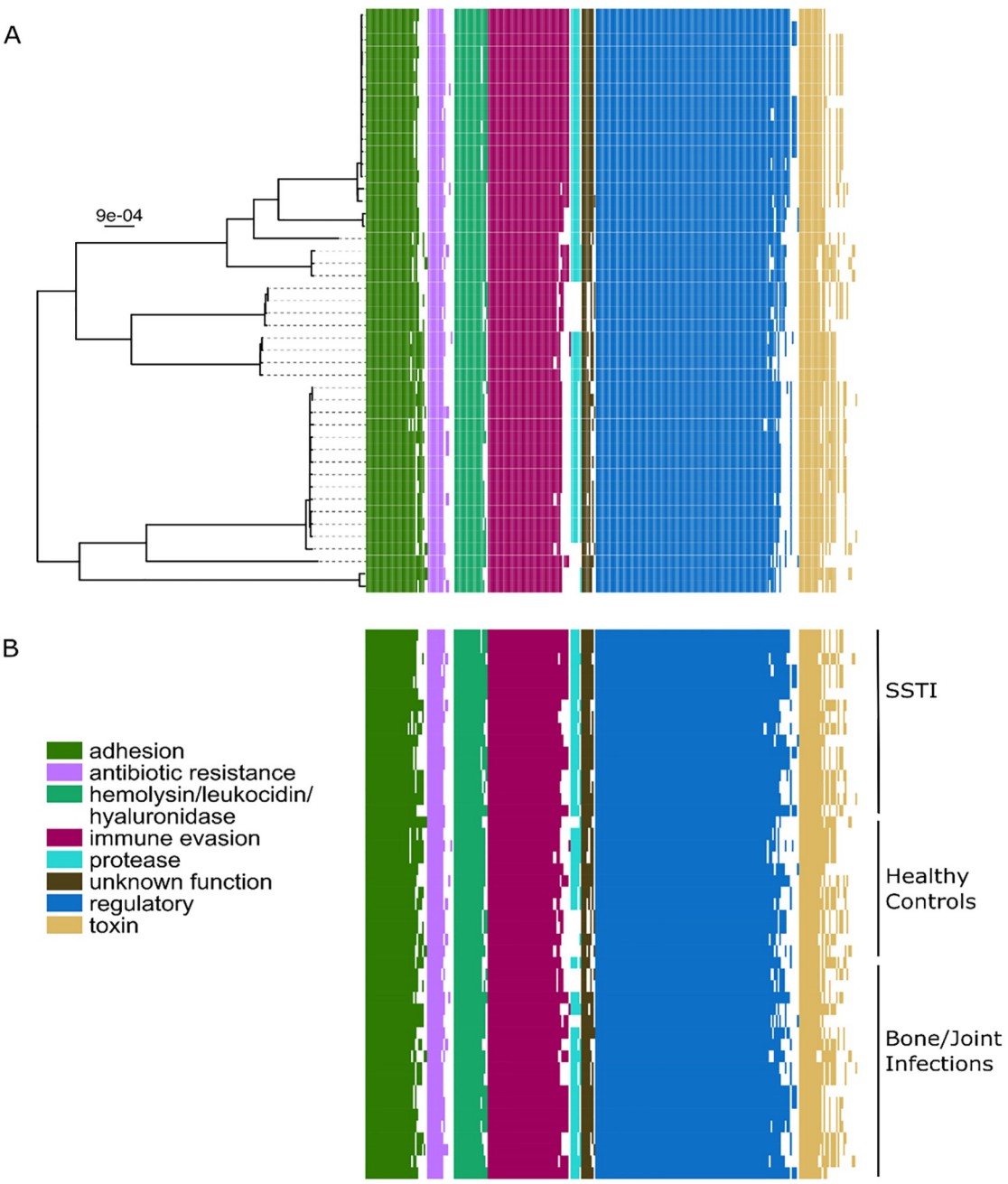

**Fig 2. Presence/absence of virulence genes in *Staphylococcus aureus* isolates from children with osteoarticular infections compared with children with skin and soft tissue abscesses and healthy controls.** A) Virulence gene presence/absence was mapped onto the core genome phylogeny for 47 *Staphylococcus aureus* sequences (phylogeny described in Fig 1). Colored boxes indicate presence of virulence genes. Virulence genes have been assigned to categories for analysis (adhesion, antibiotic resistance, etc.), which are listed in the legend. B) Virulence gene presence/absence is mapped by disease type.

healthy controls produced the most biofilm, followed by the osteoarticular isolates, and finally the skin and soft tissue infection isolates (Fig 4D). However, only healthy control isolates produced significantly more biofilm than any other category (in comparison to skin and soft tissue isolates, p = .02).

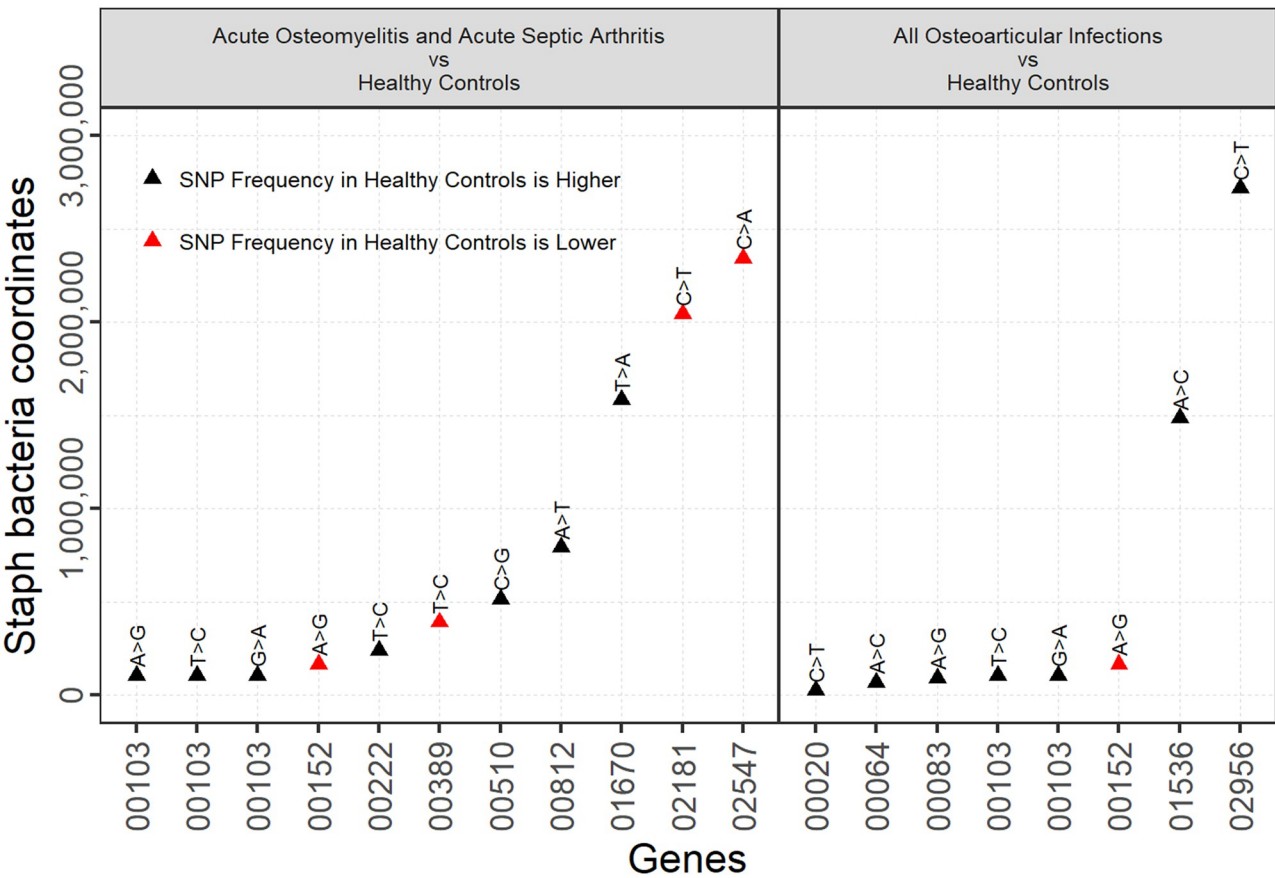

**Fig 3. Plot of virulence gene mutations in *Staphylococcus aureus* isolates from children with osteoarticular infections legend (gene function).**
00103- - -ABC transporter permease/nickel cation transport. 00152- - -isochorismatase-domain containing protein. 00222- - -TagB/teichoic acid biosynthesis. 00389- - -Superantigen-like protein. 00510- - -Serine acetyltransferase. 00812- - -clfA (clumping factor A). 01670- - -cytidine deaminase. 02181- - -PVL-like protein. 02547- - -molybdenum ABC transporter. 00020- - -WalK/WalR regulon. 00064- - -norG regulator (antibiotic efflux pump). 00083- - -sbnI (serine kinase/siderophore biosynthesis). 01536- - -sub-unit of the clp protease. 02956- - -ompR family/braB/BceR antibiotic resistance.

## Discussion

We demonstrated significantly increased carriage of genes involved in evasion of the host immune response amongst *S. aureus* isolates taken from children with osteoarticular infections compared with those involved in asymptomatic skin colonization. Interestingly, this difference was not present when compared with *S. aureus* isolates involved in the formation of skin abscesses, suggesting that the carriage of these genes may be involved in invasive disease in general. However, despite the statistical significance of this finding, the absolute difference in carriage of any immune evasion gene between the two groups was relatively small (5%). This fact, in addition to the absence of any other significant differences in virulence gene carriage noted in our data, suggests that genetic drivers of *S. aureus* osteoarticular infections may be primarily regulated by factors other than genomic content (e.g. at the transcriptional or translational level). Using a murine model of osteomyelitis, Szafranska found that 444 of 2,503 genes in *S. aureus* were differentially expressed during infection, which suggests that a greater degree of variability may be seen in children with osteomyelitis at the transcriptional level than at the genomic level [13]. In contrast to our findings, Gavira-Agudelo demonstrated slight differences in virulence gene carriage in 12 children with acute osteomyelitis due to MRSA when compared to two laboratory reference strains (76 of 78 sequenced virulence genes in children

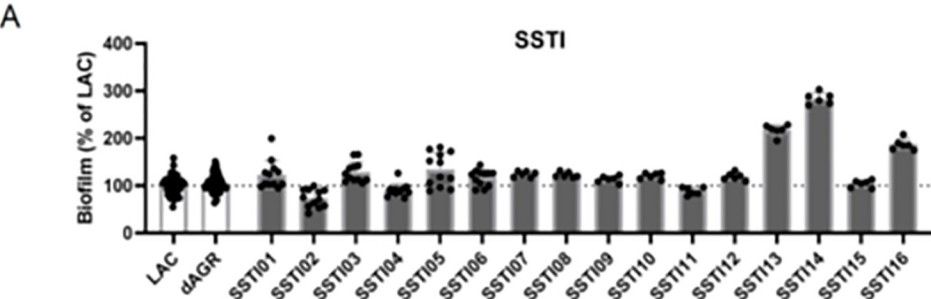

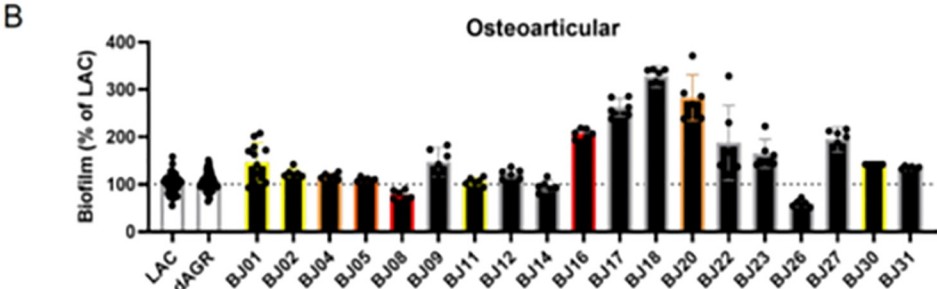

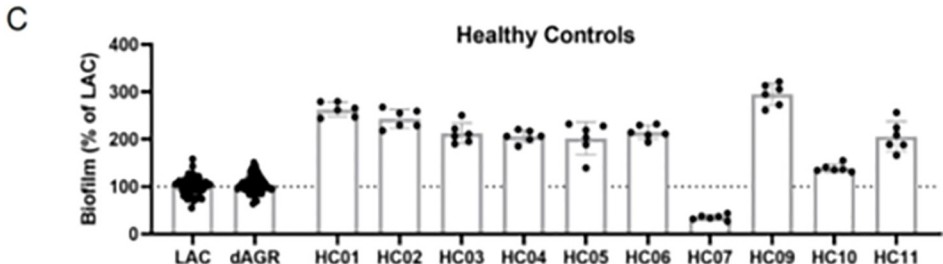

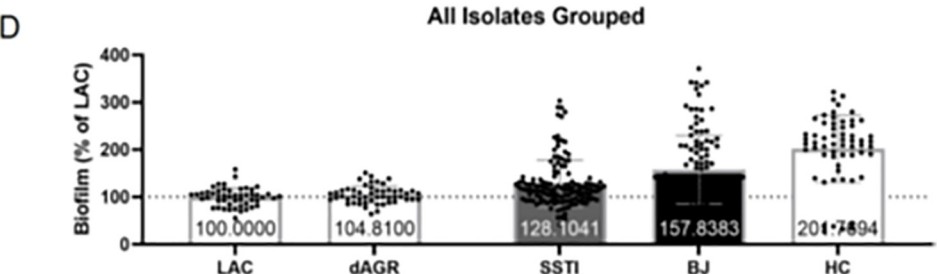

**Fig 4. Biofilm formation capacity of *Staphylococcus aureus* isolates.** The capacity for each isolate to form biofilm was assessed using a 24-hour static biofilm assay. Results are grouped by infection type with A) skin and soft tissue abscess isolates B) osteoarticular isolates and C) healthy control isolates. D) All data for each infection type were combined to generate grouped pools. Data represent the mean ± SD and are the result of at least two biological replicates with at least seven technical replicates. The USA300 LAC (AH1263) and the isogenic *agr*-deletion mutant (AH1292) were included for internal control and comparison. For grouped isolates (D), only the healthy control group demonstrated significantly more robust biofilm formation than any other group (in comparison with skin and soft tissue isolates, p = .02). SSTI = skin and soft tissue infection; BJ = bone and joint infection; HC = healthy controls.

with acute osteomyelitis vs. a mean of 62 of 78 sequenced virulence genes carried by the reference strains, p = .0124) [12]. However, as the authors discussed, the use of laboratory reference strains (isolated from older patients) as a comparator group may have biased results. In addition, this study focused only on acute osteomyelitis due to MRSA. Collins analyzed the distribution of 201 virulence genes amongst 71 children with acute osteomyelitis and found 40 genes significantly associated with severity of the osteomyelitis (as assessed by a validated clinical scoring system) [23]. These findings suggest that the genomic content of *S. aureus* may differ between more virulent and less virulent isolates. However, this study did not include isolates from control groups without osteomyelitis, and as such may not be in an optimal position to address questions of pathogenesis itself.

*Cap* gene products are involved in immune evasion of host defense [21, 40, 41], and were the most differentially distributed genes between isolates with and without osteoarticular infections in our study. However, at an individual gene level, these were not significantly different after correction for multiple comparisons. *Cap* proteins provide a measure of immune evasion which is mediated primarily through resistance to opsonization and phagocytosis [40]. Interestingly, these genes may be less frequently expressed in adults with chronic osteomyelitis than those with acute osteomyelitis [42], though the gene carriage between acute and chronic osteomyelitis did not differ in our study cohort.

Mutations in three genes regulating biofilm formation were differentially present between isolates from subjects with acute osteoarticular infections and healthy controls, the *clfA* and *clp* genes and the *WalK/R* regulon. *WalKR* expression is linked to more robust biofilm formation [43] and cellular adherence in vitro [44], *clfA* facilitates biofilm formation, bacterial attachment and colonization of surfaces [45] and *clp* (the caseinolytic or clp protease) modulates biofilm development [46]. Given the well described role of biofilm formation in the development of *S. aureus* osteoarticular infections [47], these relatively conserved genes may facilitate establishment of these infections in children. However, given the large number of *S. aureus* genes involved in biofilm formation and virulence, this represents a very small number of mutations, and as such is unlikely by itself to explain the pathogenic potential of *S. aureus* isolates involved in bone and joint infections. In addition, our phenotypic assessment of biofilm production failed to detect any differences between isolates from osteoarticular infections and other isolates. Larger studies with more power may help to identify additional genes involved in this process, and unravel the complex interplay involved between the host immune response, the transcriptional patterns of the pathogen and the impact of antimicrobial therapy on bacterial transcription over time. Interestingly, these findings were not apparent when chronic osteomyelitis was compared with control groups. However, given our small number of isolates taken from children with chronic osteomyelitis, our findings are likely to be limited in this regard.

Though a limited number of our isolates were collected serially over time or from multiple anatomic sites in our cohort, we did not notice any difference in virulence gene carriage spatially or temporally. Trouillet-Assant described whole genome sequencing of *S. aureus* isolates involved in recurrent bone and joint infections in three adults, and failed to find any significant differences in genomic content between the isolates over time [48]. Differences in the expression of hemolysin alpha, a product of the *hla* gene, were noted however [48].

ST-8, a predominant sequence type in the U.S. [49], was the most common sequence type noted in children with skin abscesses (and in those with MRSA infections) in our study, though no sequence types were specifically associated with osteoarticular infections in our cohort. However, given the predilection of *S. aureus* to evolve primarily via emergence of single nucleotide polymorphisms and horizontal gene transfer of mobile genetic elements [50], as well as the limited number of isolates in our study, our results would not be able to prove descent from genetically similar organisms but are primarily estimates of the genetic distance between isolates.

The virulence determinants of *S. aureus* osteoarticular infections are likely multi-factorial, and may involve factors such as the genomic composition of the infecting isolate [23], a necessary combination of virulence and regulatory genes [12] and transcriptional variability [14] (which in turn may be influenced by the host immune response [15], antimicrobial exposure [17] and/or the host microbiome) [18]. Hence, future study of genetic drivers of osteoarticular infections in children would ideally attempt to account for many of these factors.

Knowledge of the virulence determinants involved in *S. aureus* osteoarticular infections has direct clinical implications. Identification of genes involved in driving a particular isolate towards a predilection for osteoarticular infections (as opposed to a localized skin abscess or skin colonization) may help create novel diagnostic approaches to identify colonization with isolates exhibiting a tropism for osteoarticular sites, as well as to identify possible targets for future vaccines and immunotherapeutic agents, such as novel drugs designed to dysregulate the formation of biofilm production by *S. aureus* (e.g. via activation of the clp protease) [51].

In regards to biofilm formation, it was surprising to find that the healthy control isolates generated more robust biofilms compared with skin and soft tissue infection isolates. We expected that osteoarticular isolates would demonstrate the largest biofilm capacity, followed by isolates taken from skin and soft tissue infections, as these infection types rely on biofilms [47]. However, the large variability within each group suggests that the capacity to form biofilm is not the only requirement for each infection type. We also note that compared to our previous report [38], the *agr*-deletion mutant was only 5% above that of the isogenic LAC. The only experimental difference between the two studies was the lot numbers of the human plasma pools, suggesting that donor variability may affect biofilm assays; this would suggest that when comparing isolates a single plasma pool would be favorable to make strong and accurate conclusions.

Our study benefited from the collection of a variety of isolates from two different control groups across a spectrum of *S. aureus* involvement (asymptomatic skin colonization and skin abscesses), prospective data collection and the inclusion of a variety of osteoarticular infections due to both MRSA and MSSA. Our study was limited by a small sample size. In addition, as a single site study, it may have limited generalizability given the potential for regional differences in the clones of *S. aureus* in circulation in particular areas of the country [52]. Our use of the reference strain NCTC8325 may have limited our generalizability to currently circulating community strains. However, given the extensive annotation of the strain, the fact that it is still in use as a reference genome, and that it is familiar to investigators we think it is a valid comparator. Further, we strengthened our findings through use of internal comparisons with multiple clinical controls and phenotypic investigation and validation of noted biofilm mutations. Given the complex genetic interplay between virulence mechanisms in *S. aureus* and an ever-expanding list of known virulence genes, our list of genes was likely incomplete. However, it did provide an important first step in describing the distribution of these genes in isolates cultivated from a broad variety of infections. Further study utilizing larger sample sizes and incorporating diverse patient populations would be beneficial, as would assessments of transcriptional and translational patterns and the influence of the host immune response on transcription in children with bone and joint infections.

## Supporting information

**S1 Table. Summary statistics for *S. aureus* genome assemblies.** Assemblies were generated with short read data in Unicycler.
(DOCX)

## Acknowledgments

Special thanks to Dr. Alexander Horswill for isolates *S. aureus* AH1263 and AH1292 (University of Colorado Anschutz Medical Campus, CO, USA).

## Author Contributions

**Conceptualization:** Walter Dehority, Kylie Disch, Rebekkah Varjabedian, Parisa Mortaji, Yan Guo, Darrell Dinwiddie, Jon Femling.

**Data curation:** Walter Dehority, Daryl B. Domman, Seth M. Daly, Kathleen D. Triplett, Kylie Disch, Rebekkah Varjabedian, Aimee Yousey, Deirdre Hill, Olufunmilola Oyebamiji, Yan Guo, Pamela R. Hall, Darrell Dinwiddie.

**Formal analysis:** Walter Dehority, Valerie J. Morley, Daryl B. Domman, Seth M. Daly, Kathleen D. Triplett, Aimee Yousey, Deirdre Hill, Olufunmilola Oyebamiji, Yan Guo, Pamela R. Hall, Darrell Dinwiddie, Jon Femling.

**Funding acquisition:** Walter Dehority, Kylie Disch, Pamela R. Hall.

**Investigation:** Walter Dehority, Daryl B. Domman, Seth M. Daly, Kathleen D. Triplett, Kylie Disch, Aimee Yousey, Deirdre Hill, Olufunmilola Oyebamiji, Yan Guo, Pamela R. Hall, Darrell Dinwiddie, Jon Femling.

**Methodology:** Walter Dehority, Valerie J. Morley, Daryl B. Domman, Seth M. Daly, Kathleen D. Triplett, Aimee Yousey, Deirdre Hill, Olufunmilola Oyebamiji, Yan Guo, Kurt Schwalm, Pamela R. Hall, Darrell Dinwiddie, Jon Femling.

**Project administration:** Walter Dehority, Rebekkah Varjabedian, Parisa Mortaji, Kurt Schwalm, Pamela R. Hall, Jon Femling.

**Resources:** Seth M. Daly, Kathleen D. Triplett, Aimee Yousey, Yan Guo, Pamela R. Hall, Darrell Dinwiddie, Jon Femling.

**Software:** Daryl B. Domman, Yan Guo, Kurt Schwalm, Pamela R. Hall, Darrell Dinwiddie.

**Supervision:** Walter Dehority, Daryl B. Domman, Yan Guo, Pamela R. Hall, Darrell Dinwiddie, Jon Femling.

**Validation:** Walter Dehority, Valerie J. Morley, Seth M. Daly, Kathleen D. Triplett, Aimee Yousey, Deirdre Hill, Olufunmilola Oyebamiji, Yan Guo, Kurt Schwalm, Pamela R. Hall, Darrell Dinwiddie, Jon Femling.

**Visualization:** Walter Dehority.

**Writing – original draft:** Walter Dehority, Valerie J. Morley, Daryl B. Domman, Seth M. Daly, Kylie Disch, Rebekkah Varjabedian, Aimee Yousey, Parisa Mortaji, Deirdre Hill, Olufunmilola Oyebamiji, Yan Guo, Darrell Dinwiddie, Jon Femling.

**Writing – review & editing:** Walter Dehority, Valerie J. Morley, Daryl B. Domman, Seth M. Daly, Kathleen D. Triplett, Kylie Disch, Rebekkah Varjabedian, Aimee Yousey, Parisa Mortaji, Deirdre Hill, Olufunmilola Oyebamiji, Yan Guo, Pamela R. Hall, Darrell Dinwiddie, Jon Femling.

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
