## [Decision Letter · Decision Letter 0]

20 Dec 2021

PONE-D-21-36810Genomic characterization of Staphylococcus aureus isolates causing osteoarticular infections in otherwise healthy childrenPLOS ONE

Dear Dr. Dehority,

Thank you for submitting your manuscript to PLOS ONE. After careful consideration, we feel that it has merit but does not fully meet PLOS ONE’s publication criteria as it currently stands. Therefore, we invite you to submit a revised version of the manuscript that addresses the points raised during the review process.

Both reviewers comment on the clarity of data presentation and both suggest some additional information to enrich the content of the manuscript. Specifically, Reviewer 1 asks (1) if it is possible to correlate the Severity of Illness with the phylogeny of the pathogens? and (2) if the Discussion can include some acknowledgement of the uncertainty in assessing phylogenetic distance under the conditions of the study. Reviewer 2 asks that the authors explore the consequences of using NTCT8325 as the wild-type reference strain in the definition of a mutation, especially when literature exists that addresses the issue. Reviewer 2 also asks about the biofilm-forming ability of the strains that you have mapped if that is possible. Finally, this reviewer requests that you consider your suggestion, “genetic drivers of S. aureus osteoarticular infections may be primarily regulated by factors other than genomic content (e.g. at the transcriptional or translational level)”. If this is true, which some literature suggests, do you have data that would enrich this suggestion? Or is there information in the literature that would bridge your data and this presumed regulation?

We look forward to receiving your revised manuscript.

Kind regards,

Noreen J. Hickok, Ph.D.

Academic Editor

PLOS ONE

Journal Requirements:

a) Did participants provide their written or verbal informed consent to participate in this study?

Reviewers' comments:

Reviewer's Responses to Questions

**Comments to the Author**

1. Is the manuscript technically sound, and do the data support the conclusions?

Reviewer #1: Yes

Reviewer #2: Partly

2. Has the statistical analysis been performed appropriately and rigorously? 

Reviewer #1: Yes

Reviewer #2: Yes

3. Have the authors made all data underlying the findings in their manuscript fully available?

Reviewer #1: Yes

Reviewer #2: Yes

4. Is the manuscript presented in an intelligible fashion and written in standard English?

Reviewer #1: Yes

Reviewer #2: Yes

5. Review Comments to the Author

Reviewer #1: This is an interesting and well-written manuscript. The study methodology is sound and the conclusions are supported by the data. I feel that this manuscript is suitable for publication within PLOS ONE as it satisfies the criteria for PLOS ONE research and will make an important contribution to the scientific record in the field of Staph aureus disease pathogenesis and genetic mechanisms.

There are a few minor revision considerations which I would propose to improve this study. I think the distinction between various forms of osteoarticular infection which the authors intend may not be as meaningful as suggested. Chronic osteomyelitis is often the outcome of inadequate recognition and treatment of acute hematogenous osteomyelitis. Septic arthritis due to Staph aureus in older children is more likely to be contiguous to an unrecognized adjacent focus of osteomyelitis than to represent primary septic arthritis (which usually occurs in younger children and from a variety of pathogens). While Staph aureus may cause primary septic arthritis, I am skeptical whenever the pathogen is S.aureus, the child is older (median 8 years) and in the presence of bacteremia.

A more meaningful, and somewhat objective, way to explore clinical differences between S. aureus isolates that led to invasive infection of bone (and/or joint) is the calculation of Severity of Illness from readily available clinical and laboratory parameters which are identified during the first few days of hospitalization. It would be very interesting to see how clinical severity of illness maps out according to the phylogenetic relationship of the OAI pathogens in this study.

The authors finding that generally the bacteria have a robust genetic armamentarium, regardless of whether they were isolated from asymptomatic carriers, SSTIs, or OAI. This fits my intuition that the difference in clinical progression is likely to be identified in the transcriptome and proteotome. This is the next area of inquiry, inevitably.

A commendable part of this study is the simplicity of design including colonization pathogens, skin infection pathogens, and deep infection pathogens. This is missing from the previous publications of the genomic heterogeneity of S. aureus clinical isolates in a single community.

Staph aureus does not have a lot of major genetic recombinations or mutations. It most likely evolves through single nucleotide polymorphisms and lateral transfer of small amounts of genetic material through plasmids. Phylogenetic analysis is therefore challenging as the analysis performed over 3 ½ years of enrollment for a small number of isolates is only a mathematical estimation of genetic distance between isolates and does not imply phylogeny or descent from related pathogens. That should be acknowledged in the discussion.

Reviewer #2: The manuscript by Dehority et al describes genome scale studies aimed at determining whether there are specific genetic determinants encoded by Staphylococcus aureus that can be correlated with the capacity to cause invasive osteoarticular infections in children. S. aureus is a unique bacterial pathogen in its capacity to cause infection in essentially any human tissue, and it is very clear that it is the leading cause of bone and joint infection. For these reasons, there has been a great deal of interest in assessing the possibility that correlations exist between the presence of specific virulence factors and the capacity to cause specific forms of infection. Many studies have been published that attempted to address this issue, and given the predominance of S. aureus many of these have focused on bone and joint infections. None of these have reached definitive conclusions about the presence or absence of specific genes, and this manuscript is no exception. Thus, it could be argued that this manuscript adds little to the existing literature, particularly given the relatively small cohort of patients/isolates examined. However, in my view, the study was very well done and included appropriate exclusion criteria to identify isolates associated with invasive osteoarticular infection without obvious risk factors (n = 19) and controls in the form of both colonization isolates (n = 12) and isolates from “superficial” skin and soft tissue infections (n = 16). The manuscript is also very well written, and the authors appropriately note the limitation of their conclusions. The results indicated a significant correlation between the presence of immune evasion genes in isolates from osteoarticular infection vs. colonization controls, but this distinction was not present between isolates from osteoarticular vs. skin infection. There was no identifiable correlation with specific clonal lineages, although ST8 and ST30 did appear to be predominant among the isolates included in this study irrespective of infection type. The authors did identify mutations in 14 genes involved in biofilm formation, but as with the immune evasion genes the distinction was between isolates from osteoarticular infection and the colonization controls but not between osteoarticular isolates and isolates from skin infection. Additionally, these “mutations” appear to have been defined by comparison to the NCTC8325 genome, and there is no description of how they were defined as mutations beyond that. This assumes that NCTC8325, which is NOT a contemporary clinical isolate, can be defined as a wild-type strain, and there is considerable data in the literature to suggest that this is not the case. In this regard, skin infections are also considered biofilm associated infections, so it would have been of interest if the authors had examined these correlations at a phenotypic level (i.e. did the colonization strains and infection strains have a significant difference in their capacity to form a biofilm?). Similarly, the primary conclusion from the manuscript is that “genetic drivers of S. aureus osteoarticular infections may be primarily regulated by factors other than genomic content (e.g. at the transcriptional or translational level)”, and I believe the manuscript would be significantly enhanced if this possibility had been examined. Indeed, the authors even note a previous manuscript that found differences in the expression of hla, which along with protein A is generally considered a strong indicator differences in regulatory circuits in S. aureus and, like biofilm formation, is an easy phenotype to assess.

6. PLOS authors have the option to publish the peer review history of their article (what does this mean?). If published, this will include your full peer review and any attached files.

Reviewer #1: **Yes: **Lawson A. Copley, MD, MBA

Reviewer #2: No

---

## [Author Response · Author response to Decision Letter 0]

13 Jun 2022

Response to Reviewers

Note that we have only presented the questions and specific requests from the overall comments from the reviewers in an effort to simplify the response letter. 

Reviewer #1

1.) A more meaningful, and somewhat objective, way to explore clinical differences between S. aureus isolates that led to invasive infection of bone (and/or joint) is the calculation of ‘Severity of Illness’ from readily available clinical and laboratory parameters which are identified during the first few days of hospitalization. It would be very interesting to see how clinical severity of illness maps out according to the phylogenetic relationship of the OAI pathogens in this study.

Response: An excellent suggestion, and we were able to do that. We used the scoring system proposed by Athey, et al in 2019 (Athey, et al. J Pediatr Orthop. 2019;39:90-97. PMID 27741035), which was a slight modification of a prior scoring system (Copley et al. Pediatr Infect Dis J. 2014;33:35-41. PMID 24352188) to categorize all subjects with acute osteomyelitis in our study. Severity scores ranged from 1 to 5. These scores were incorporated into the phylogeny tree in Figure 1, with updates to the figure legend. Additional methods verbiage was added to lines 151-157 as well as descriptions in the results section (lines 247-248) and the legend to Figure 1 regarding these additions. Note that we also re-generated the phylogeny to get the bootstrap support values for Figure 1. The regenerated tree in Figure 2 has some minor cosmetic changes relative to the original submission (e.g. rotated nodes). We also recreated Figure 2 so that tree structures are identical in both figures. 

2.) Staph aureus does not have a lot of major genetic recombinations or mutations. It most likely evolves through single nucleotide polymorphisms and lateral transfer of small amounts of genetic material through plasmids. Phylogenetic analysis is therefore challenging as the analysis performed over 3 ½ years of enrollment for a small number of isolates is only a mathematical estimation of genetic distance between isolates and does not imply phylogeny or descent from related pathogens. That should be acknowledged in the discussion.

Response: We have added denotations indicating nodes with ultrafast bootstrap support of at least 95% in Figure 1, which provides information on uncertainty in the presented phylogenetic relationships, and referenced this in the legend to Figure 1. We have also added verbiage to this effect in the discussion section, lines 428-432, as well as a new reference to this effect (#50). 

Reviewer #2

1.) Additionally, these “mutations” appear to have been defined by comparison to the NCTC8325 genome, and there is no description of how they were defined as mutations beyond that. This assumes that NCTC8325, which is NOT a contemporary clinical isolate, can be defined as a wild-type strain, and there is considerable data in the literature to suggest that this is not the case. In this regard, skin infections are also considered biofilm associated infections, so it would have been of interest if the authors had examined these correlations at a phenotypic level (i.e. did the colonization strains and infection strains have a significant difference in their capacity to form a biofilm?).

Response: An excellent point. This is a limitation of the study, and, as you correctly note, one that is discussed in the literature. However, given the extensive annotation of the NCTC8325 strain, the fact that it is still in use as a reference genome, and that it is familiar to investigators we think it still has validity as a comparator. Further, our use of internal comparisons with multiple clinical controls and the phenotypic investigation/validation of the noted biofilm mutations (suggested by you and discussed below), would suggest that our findings have merit. We would also note that the new phenotypic biofilm assessments, as well as the extensive phylogenetic analyses (including the newly added stratification by severity of illness) coupled with the use of our multiple internal control groups are all novel approaches in this subject population. We have added verbiage to this effect to the discussion section, in lines 461-466.

To attempt to address this issue further, and to incorporate your biofilm suggestions, we performed analyses to assess capacity to form biofilm with the isolates. Interestingly, no difference existed in the ability of the infection-associated isolates to form biofilm, though isolates taken from healthy controls were significantly more robust in this regard. New verbiage addressing the biofilm assays has been added to lines 37-38 and 44-45 in the abstract, in the methods section in lines 158-173 (describing the experimental approach) and lines 189-195 (describing the statistical approach to analyzing results), in the results section (lines 347-359), in the discussion section (lines 412-414 and lines 446-455) and in Figure 4. We have also added Seth Daly, Kathleen Triplett and Pamela Hall (and acknowledged the NIH funding of Dr. Hall) to the manuscript, as they assisted with the biofilm assays. 

2.) Similarly, the primary conclusion from the manuscript is that “genetic drivers of S. aureus osteoarticular infections may be primarily regulated by factors other than genomic content (e.g. at the transcriptional or translational level)”, and I believe the manuscript would be significantly enhanced if this possibility had been examined.

Response: We agree with this viewpoint. We discussed this option extensively as the study was being planned, but we felt we did not have the resources or finances to pursue transcriptional or translational analyses at the time, and as a result, focused on genomic analyses. Following this study, however, we would like to incorporate these into future work.

---

## [Decision Letter · Decision Letter 1]

20 Jul 2022

Genomic characterization of Staphylococcus aureus isolates causing osteoarticular infections in otherwise healthy children

PONE-D-21-36810R1

Dear Dr. Dehority,

We’re pleased to inform you that your manuscript has been judged scientifically suitable for publication and will be formally accepted for publication once it meets all outstanding technical requirements.

Kind regards,

Noreen J. Hickok, Ph.D.

Academic Editor

PLOS ONE

Additional Editor Comments (optional):

Reviewers' comments:

Reviewer's Responses to Questions

**Comments to the Author**

1. If the authors have adequately addressed your comments raised in a previous round of review and you feel that this manuscript is now acceptable for publication, you may indicate that here to bypass the “Comments to the Author” section, enter your conflict of interest statement in the “Confidential to Editor” section, and submit your "Accept" recommendation.

Reviewer #1: All comments have been addressed

Reviewer #2: All comments have been addressed

2. Is the manuscript technically sound, and do the data support the conclusions?

Reviewer #1: Yes

Reviewer #2: Yes

3. Has the statistical analysis been performed appropriately and rigorously? 

Reviewer #1: Yes

Reviewer #2: Yes

4. Have the authors made all data underlying the findings in their manuscript fully available?

Reviewer #1: Yes

Reviewer #2: Yes

5. Is the manuscript presented in an intelligible fashion and written in standard English?

Reviewer #1: Yes

Reviewer #2: Yes

6. Review Comments to the Author

Reviewer #1: All concerns have been addressed. No additional revision is required from my standpoint. I appreciate the opportunity to review this research.

Reviewer #2: The authors report data investigating the possibility that genomic differences exist that define isolates of S. aureus that cause osteoarticular infections in children without underlying risk factors. This is a challenging task, and it could be argued that any significant correlations would require evaluation of a much larger set of isolates. That said, the authors compared 47 isolates more or less equally distributed between "uninfected" colonizers, isolates from patients with skin abscesses, and isolates with osteoarticular infections. I believe this number is sufficient to support their conclusions, and they have address all of the concerns expressed in the previous review.

7. PLOS authors have the option to publish the peer review history of their article (what does this mean?). If published, this will include your full peer review and any attached files.

Reviewer #1: No

Reviewer #2: No

---

## [Editor Report · Acceptance letter]

19 Aug 2022

PONE-D-21-36810R1 

Genomic characterization of *Staphylococcus aureus* isolates causing osteoarticular infections in otherwise healthy children 

Dear Dr. Dehority:

I'm pleased to inform you that your manuscript has been deemed suitable for publication in PLOS ONE. Congratulations! Your manuscript is now with our production department. 

Kind regards, 

on behalf of

Dr. Noreen J. Hickok 

Academic Editor

PLOS ONE